# Carbapenem-Resistant *Acinetobacter baumannii*: Biofilm-Associated Genes, Biofilm-Eradication Potential of Disinfectants, and Biofilm-Inhibitory Effects of Selenium Nanoparticles

**DOI:** 10.3390/microorganisms11010171

**Published:** 2023-01-10

**Authors:** Aleksandra Smitran, Bojana Lukovic, LJiljana Bozic, Dijana Jelic, Milos Jovicevic, Jovana Kabic, Dusan Kekic, Jovana Ranin, Natasa Opavski, Ina Gajic

**Affiliations:** 1Faculty of Medicine, University of Banja Luka, 78 000 Banja Luka, Bosnia and Herzegovina; 2Academy of Applied Studies Belgrade, College of Health Sciences, 11000 Belgrade, Serbia; 3Department of Chemistry, Faculty of Natural Sciences and Mathematics, University of Banja Luka, 78 000 Banja Luka, Bosnia and Herzegovina; 4Institute of Microbiology and Immunology, Faculty of Medicine, University of Belgrade, 11000 Belgrade, Serbia

**Keywords:** carbapenem-resistant *Acinetobacter baumannii*, disinfectants, biofilm-associated genes, selenium nanoparticles

## Abstract

This study aimed to investigate the biofilm-production ability of carbapenem-resistant *Acinetobacter baumannii* (CRAB), the biofilm-eradication potential of 70% ethanol and 0.5% sodium hypochlorite, the effects of selenium nanoparticles (SeNPs) against planktonic and biofilm-embedded CRAB, and the relationship between biofilm production and bacterial genotypes. A total of 111 CRAB isolates were tested for antimicrobial susceptibility, biofilm formation, presence of the genes encoding carbapenemases, and biofilm-associated virulence factors. The antibiofilm effects of disinfectants and SeNPs against CRAB isolates were also tested. The vast majority of the tested isolates were biofilm producers (91.9%). The *bap*, *ompA*, and *csuE* genes were found in 57%, 70%, and 76% of the CRAB isolates, with the *csuE* being significantly more common among biofilm producers (78.6%) compared to non-biofilm-producing CRAB (25%). The tested disinfectants showed a better antibiofilm effect on moderate and strong biofilm producers than on weak producers (*p* < 0.01). The SeNPs showed an inhibitory effect against all tested planktonic (MIC range: 0.00015 to >1.25 mg/mL) and biofilm-embedded CRAB, with a minimum biofilm inhibitory concentration of less than 0.15 mg/mL for 90% of biofilm producers. In conclusion, SeNPs might be used as promising therapeutic and medical device coating agents, thus serving as an alternative approach for the prevention of biofilm-related infections.

## 1. Introduction

*Acinetobacter baumannii* is an important opportunistic pathogen that causes a variety of infections in hospital settings, particularly in critically ill patients with predisposing risk factors such as an intensive care unit stay, assisted ventilation, surgical interventions, invasive procedures, and the presence of indwelling devices (catheters, drainage or endotracheal tubes, and artificial implants) [1,2]. Factors that contribute to an increasing trend of *A. baumannii* nosocomial infections worldwide sum up its persistence and growth in medical wards due to resistance to desiccation, disinfectants, and antibiotics, and ability to form biofilm on biotic and abiotic surfaces [1]. Until recently, carbapenems were the standard therapy for treating *A. baumannii* nosocomial infections. However, the spread of carbapenem-resistant *A. baumannii* (CRAB) strains, which typically exhibit a multidrug-resistant (MDR) phenotype, raises severe concerns about treatment options [3].

As a result, *A. baumannii* is one of the ESKAPE organisms, a group of clinically important pathogens with the potential to develop significant levels of antimicrobial resistance, while CRAB is listed on the World Health Organization’s priority list of antibiotic-resistant bacteria for effective drug development [4,5]. Carbapenem resistance in *A. baumannii* is mostly associated with the production of carbapenem-hydrolysing enzymes, Ambler class D β-*lactamases,* or oxacillinases (OXAs) [6]. The high prevalence of CRAB carrying *bla*_OXA-23-like_ and *bla*_OXA-24-like_ genes has emerged as a serious problem in healthcare settings in Serbia [7,8].

In addition to antimicrobial resistance, an important issue with *A. baumannii* is the biofilm formation rate, which is higher than in other species [1]. Several virulence factors are involved in the biofilm formation in *A. baumannii,* such as outer membrane protein A (OmpA), biofilm-associated protein (bap), and pili. OmpA is the best-studied highly stable outer membrane protein known to have a critical role in regulating *A. baumannii* adhesion and biofilm formation [9]. Bap is a high molecular weight surface protein involved in the bacterial adhesion process, a crucial step in initiating infection and starting biofilm production [10]. Csu pili, the most prevalent pili among bacteria, also have an important role in cell adhesion and biofilm formation [11].

Biofilm production can lead to chronic infections and provide additional mechanisms for antimicrobial resistance. All factors mentioned above are the main justifications for studying biofilm eradication or the prevention of biofilm development in CRAB isolates. Biofilm formed on medical devices or inanimate surfaces in hospitals is routinely exposed to disinfectants. Two of them, 0.5% sodium hypochlorite (NaOCl) and 70% ethanol, are frequently used. Since disinfectants are often unable to completely eliminate MDR germs from abiotic hospital surfaces, many scientists are striving to develop novel antimicrobial materials, such as nanoparticles and nanocomposites, to combat hospital infections. Selenium nanoparticles (SeNPs) provide a reduced risk of selenium toxicity and show excellent antimicrobial efficacy against various planktonic and biofilm-embedded pathogens [12]. However, there are scarce data about SeNPs’ activity against planktonic MDR *A. baumannii,* their inhibitory effect on biofilm formation, or biofilm-embedded MDR *A. baumannii* [13,14]. Therefore, the aims of the present study were the following: (i) to estimate the ability of CRAB isolates to form biofilm, (ii) to evaluate the eradication potential of 70% ethanol and 0.5% NaOCl on the mature biofilm, (iii) to assess the inhibitory effects of SeNPs against planktonic CRAB and biofilm formation, and (iv) to analyse the relationships between biofilm formation, biofilm-associated genes, and antimicrobial resistance.

## 2. Materials and Methods

### 2.1. Bacterial Isolates and Species Identification

The present multicentre study included 117 non-redundant randomly selected *A. baumannii* isolates from the bacterial collection of the Institute of Microbiology and Immunology in Belgrade. The isolates were recovered from patients admitted to six secondary and tertiary care hospitals in Serbia, Montenegro, and Bosnia and Herzegovina in 2017 and were part of the bacterial collection partially described in our previously published study [15]. The identification of *A. baumannii* was carried out by detecting the intrinsic *bla*_OXA-51-like_ carbapenemase gene and *rpoB* gene sequence analysis, as previously described [16,17]. An invasive isolate was defined as that obtained from normally sterile body sites (e.g., blood, peritoneal fluid, and cerebrospinal fluid). Isolates from skin and soft tissue, urine, and respiratory tract specimens were considered non-invasive.

### 2.2. Antimicrobial Susceptibility Testing

According to the recommendations of the Clinical and Laboratory Standards Institute (CLSI), the antimicrobial susceptibility of *A. baumannii* to ampicillin-sulbactam, piperacillin-tazobactam, ceftriaxone, cefotaxime, ceftazidime, cefepime, meropenem, imipenem, gentamicin, amikacin, tobramycin, tetracycline, ciprofloxacin, levofloxacin, and trimethoprim-sulfamethoxazole was determined using a disk–diffusion assay (Bio-Rad, Watford， UK) [18]. A ComASP Colistin (Liofilchem, Roseto degli Abruzzi, Italy) and Gradient strip test (Liofilchem, Roseto degli Abruzzi, Italy) were used to determine the minimum inhibitory concentrations (MICs) for colistin, imipenem, and meropenem, respectively [18]. MICs for tigecycline were determined using the broth microdilution method with Mueller–Hinton (MH) broth (Bio-Rad, Watford, UK) [18]. Except for tigecycline, whose susceptibility categories were interpreted following the European Committee on Antimicrobial Susceptibility Testing (EUCAST) breakpoints for Enterobacterales (S ≤ 0.5 μg/mL; R > 0.5 μg/mL) [19], all antibiotic susceptibility categories were determined in accordance with the CLSI criteria [18]. Only *A. baumannii* resistant to carbapenems were subjected to further analysis. All CRAB isolates were classified as follows: multidrug-resistant (MDR) (resistant to at least one agent in three or more antimicrobial categories), extensively drug-resistant (XDR) (resistant to at least one agent in all but two or fewer antimicrobial categories) and pandrug-resistant (PDR) (resistant to all agents in all antimicrobial categories tested) [20].

#### Detection of Carbapenemase-Encoding Genes

The DNA extraction of the tested isolates was carried out using a QIAamp DNA Mini Kit (QIAGEN GmbH, Hilden, Germany), according to the manufacturer’s instructions. All CRAB isolates were screened for the presence of *bla*_OXA-23-like_, *bla*_OXA-24-like_, *bla*_OXA-58-like_, and *bla*_OXA-143-like_ with primers published elsewhere (Appendix A) [21,22].

### 2.3. Biofilm Production Assay

The quantification of biofilm biomass was carried out following the protocol of Stepanovic et al. [23]. Briefly, 100 μL of *A. baumannii* suspension in Trypticase soy broth (TS, Biorad, UK) with a final concentration of 10^6^ CFU/mL was transferred to each well of a 96-well microtiter plate and incubated for 24 h at 37 °C. Biofilm growing in the microtiter plate was dyed with 100 μL of 2% (*w*/*v*) crystal violet for 15 min. Glacial acetic acid at 33% (*v*/*v*) was used to dissolve the dye bound to the adhering cells inside the biofilm matrix. Using an automated microtiter plate reader, each well’s optical density (OD) was determined at 570 nm (ICN Flow Titertek Multiskan Plus Reader, Meckenheim, Germany). TS broth was the only suspension in the negative control wells. As the positive control, *A. baumannii* ATCC 19606 was used. Three standard deviations more than the mean OD of the negative control were designated as the cut-off OD (ODc).

The results were evaluated as follows:

OD ≤ ODc non-biofilm producers, ODc < OD ≤ (2 × ODc) = weak biofilm producers, (2 × ODc) < OD ≤ (4 × ODc) = moderate biofilm producers, and OD > (4 × ODc) = strong biofilm producers.

### 2.4. Genes Encoding Virulence Factors Associated with Biofilms

A total of 74/111 randomly selected CRAB strains were screened for the presence of the following biofilm-related genes: *bap* (encoding the biofilm-associated protein), *csuE* (encoding the csu pilus protein subunit), and *ompA* (encoding the outer membrane protein A). All genes were amplified through PCR using the Qiagen Taq PCR Master Mix Kit (Qiagen, Hilden, Germany), according to the manufacturer’s instructions. A list of primers, their sequences, annealing temperatures, and sizes of the amplified products used in this study is shown in Appendix A, Table A1.

### 2.5. Antibiofilm Effect of 70% Ethanol and 0.5% NaOCl on Mature Biofilm

The antibiofilm effect of disinfectants was measured according to the protocol of Narrayanan et al. [24]. CRAB clinical isolates in a final concentration of 10^6^ CFU/mL were incubated overnight in 100 μL of TS broth at 37 °C. Following biofilm formation, the microtiter plates were washed and dried at room temperature for 4 h. Then, the antibiofilm effect of 100 μL of each disinfectant (70% ethanol and 0.5% NaOCl) was tested with an exposure time of 5 and 10 min. Afterwards, the microtiter plates were washed and dried overnight at room temperature. Biofilm production of the treated isolates was measured according to the method of Stepanovic et al. [25].

### 2.6. Selenium Nanoparticles Synthesis

Two solutions of Na_2_SeO_3_ and cetyltrimethylammonium bromide (CTAB) were prepared to contain 0.5 g of Na_2_SeO_3_ and 0.25 g of CTAB. Both solutions were placed in a UV bath for 10 min and afterwards were mixed and stirred with a magnetic stirrer for 10 min. Upon stirring, 1.5 g of vitamin C was added, and the solution was heated for 4 h at 80 °C. Upon heating, 6.25 mL H_2_O and 1.875 mL of hydrazine were added into the solution and heated for 2 h at the same temperature. After ageing for four days at room temperature, the solution was filtered and washed with distilled water and dried in an oven for 1 h at 50 °C.

### 2.7. Antimicrobial Testing of Selenium Nanoparticles against Planktonic and Biofilm-Embedded Carbapenem-Resistant Acinetobacter baumannii Isolates

The MICs of SeNPs on planktonic CRAB isolates were determined using the broth microdilution method (BMD), according to the CLSI recommendations [18]. Twofold serial dilutions of SeNPs were made using MH broth (Biorad, Watford, UK). Each well in the microtiter plates was filled with 100 µL of MH broth with SeNPs (final concentration ranging from 1.25–0.0015 mg/mL) and 5 µL of bacterial inoculum (concentration of 10^7^ CFU/mL) to achieve a final concentration of 5 × 10^5^ CFU/mL. The microtiter plates were incubated for 16–18 h at 37 °C. The MIC was defined as the lowest concentration which inhibited growth detected by the unaided eye.

The minimum biofilm inhibitory concentration (MBIC) of SeNPs was determined following the protocol of Bagheri-Josheghani et al. [26], with a slight modification of the broth used in the experiment. Aliquots of 10 µL of CRAB isolates (concentration of 10^8^ CFU/mL) were incubated overnight at 37 °C in microtiter plates containing 100 µL of doubled-strength MH broth and 100 µL of SeNPs (concentrations ranging from 1.25–0.0015 mg/mL). Afterwards, the microtiter plates were washed three times with PBS, dried at room temperature and dyed with 100 μL of 2% (*w*/*v*) crystal violet for 15 min. Biofilm production of treated isolates was measured according to the method of Stepanovic et al. [25].

### 2.8. Statistical Analysis

SPSS version 20.0 (SPSS Inc., Chicago, IL, USA) was used for statistical analysis. Categorical variables are presented as absolute frequencies and percentages (%). Statistical comparisons of multiple groups with normal data distribution were carried out using one-way analysis of variance (ANOVA), followed by the Tukey post-hoc test. Statistical comparison of multiple groups on the occasion of deviation of normal data distribution was carried out using the Kruskal–Wallis rank sum test or Wilcoxon’s signed rank test. To compare variances without normal distribution between the two groups, the Mann–Whitney U test was used. Fisher’s exact test was used for comparison of the frequency of occurrence of the analysed categorical variables. A *p* value less than 0.05 was considered to be significant.

The study was approved by the ethical committee of the Medical Faculty, University of Belgrade (permission number No. 1550/IX-16).

## 3. Results

### 3.1. Bacterial Isolates and Antimicrobial Susceptibilities

Overall, 111 out of 117 *A. baumannii* included in the study were CRAB (94.9%). The antimicrobial resistance rates of CRAB isolates were the following: ampicillin-sulbactam, 62.3%; piperacillin/tazobactam, 100%; ceftriaxone, 100%; cefotaxime, 100%; ceftazidime, 100%; cefepime, 100%; ciprofloxacin, 97.2%; levofloxacin 96.8%; gentamicin, 96.7%; amikacin, 94.6%; tobramycin, 80.0%; tetracycline, 96.9%; trimethoprim/sulfamethoxazole, 93.8%; tigecycline, 25.6%; and colistin, 2.7%. Among CRAB isolates, 26.1% (*n* = 29), 71.2% (*n* = 79), and 2.7% (*n* = 3) were MDR, XDR, and PDR, respectively.

CRAB isolates were recovered from the lower respiratory tract specimens (*n* = 38, 34.2%), skin and soft tissue (*n* = 26, 23.4%), blood (*n* = 35, 31.5%), peritoneal fluid (*n* = 8, 7.2%), urine (*n* = 3, 2.7%), and cerebrospinal fluid (*n* = 1, 0.9%). The median age of the patients with infections caused by CRAB isolates was 64 years, with a range of 22–85 years.

#### Carbapenemase-Encoding Genes

All tested isolates harboured the naturally occurring *bla*_OXA-51_-like gene and the corresponding sequence of the *rpoB* gene and were, therefore, included in the study as *A. baumannii*. Of 111 CRAB isolates, 32 (28.8%) carried *bla*_OXA-24-like,_ and 39 (35.1%) carried *bla*_OXA-23-like_ genes. However, *bla*_OXA-58-like_ and *bla*_OXA-143-like_ genes were not detected in the tested population.

### 3.2. Biofilm Production Assay

Overall, 102 out of 111 CRAB isolates (91.9%) were biofilm producers. The distribution of the biofilm-production abilities of the tested isolates obtained from various clinical samples is displayed in Table 1.

All nine non-biofilm-producing isolates were non-invasive CRAB. However, a higher percentage of invasive isolates (*n* = 16, 36.4%) were classified as strong biofilm producers compared to non-invasive isolates (*n* = 22, 32.8%), *p* > 0.05.

There were no significant differences in the biofilm-production ability between MDR and XDR CRAB isolates (93.1% vs. 87.3%, *p* > 0.05).

### 3.3. Genes Encoding Virulence Factors Associated with Biofilms

Out of 111 isolates, 74 were randomly chosen and tested for the presence of the *bap*, *ompA*, and *csuE* genes. Among them, 70 were biofilm producers and four isolates were non-biofilm-producing CRAB. Overall, *bap*, *ompA*, and *csuE* genes were found in 57%, 70%, and 76% of the tested isolates, respectively. The detailed distribution of the tested biofilm-associated genes is shown in Figure 1. Among non-biofilm producers, bap and OmpA surface protein-encoding genes were equally distributed (50%), while the *csuE* was less common (25%), as illustrated in Figure 1. All biofilm-producing isolates harboured at least one of the tested genes associated with biofilms. In contrast, the presence of the *csuE* gene was more common among biofilm producers (55/70) (*p* < 0.05), while there was no statistically significant difference between defined biofilm-producing groups. In our study, *bla*_OXA-24-like_ was more commonly found among non-biofilm producing CRAB strains (77.8%) compared to biofilm producers (20.6%), while *bla*_OXA-23-like_ was evenly distributed among non- and biofilm-producing strains: 22.2% and 28.4%, respectively (*p* > 0.05).

### 3.4. Antibiofilm Effect of 70% Ethanol and 0.5% NaOCl on Mature Biofilm

In this study, we evaluated the antibiofilm effect of two disinfectants frequently used in hospitals. Both bleach and ethanol displayed antibiofilm effects in all biofilm-producing isolates. Bleach completely eradicated the biofilm of 90/111 and 109/111 isolates after 5 and 10 min of exposure, respectively. On the other hand, ethanol completely eradicated the biofilm of only 18/111 and 29/111 isolates after an exposure time of 5 and 10 min, respectively. In other isolates, ethanol was only effective in reducing preformed biofilm mass after 5 and 10 min treatments (Figure 2).

In moderate and strong biofilm producers, 70% ethanol and 0.5% bleach showed strong antibiofilm effects (*p* < 0.01). For both exposure times, 0.5% bleach was able to eradicate biofilm, which is depicted in Figure 2, since mean ODs are equal or lower compared to the ODs of the non-biofilm-forming CRAB isolates. On the other hand, 70% ethanol was not able to completely eradicate biofilm mass, but rather significantly reduced it (*p* < 0.01).

### 3.5. Selenium Nanoparticles and Their Antimicrobial Effects on Carbapenem-Resistant Acinetobacter baumannii Isolates

#### 3.5.1. Selenium Nanoparticles Characterisation

Characterisation of SeNPs was performed using a scanning electron microscopy (SEM)/energy-dispersive X-ray spectroscopy (EDX) technique. Images were produced using TM3030 SEM Hitachi Japan SEM. Figure 3 presents the SEM micrographs (left) of the synthesised SeNPs, which showed well-defined spherical morphology. The EDX spectrum (right) contains a pronounced Se peak, along with the nitrogen (N), carbon (C), and oxygen (O) peaks, which could originate from the cationic surfactant structure of CTAB, used for the production of SeNPs. As Figure 3 clearly shows, there is no agglomeration process, which could be attributed to the presence of CTAB. Its role is to act as a capping agent and to preserve the stability of highly energetic nanoparticles, likely by inducing the hydrophobicity of the SeNPs.

#### 3.5.2. Antimicrobial Effects of Selenium Nanoparticles against Planktonic and Biofilm-Embedded Carbapenem-Resistant *Acinetobacter baumannii* Isolates

SeNPs showed an antibacterial effect on all planktonic CRAB isolates and biofilm-inhibitory activity against 103 (92.7%) biofilm-producing isolates. The SeNP MIC values for planktonic isolates ranged from 0.0015 to >1.25 mg/mL, whereas MIC_50_ and MIC_90_ were 0.03 and 0.15, respectively. The median MIC value for all tested isolates was 0.03 mg/mL. There was no statistically significant difference in MIC values of SeNP between invasive and non-invasive CRAB strains (*p* > 0.05).

Additionally, SeNPs showed biofilm-inhibition activity on 105 biofilm-producing CRAB, with only two isolates with MBIC above 1.25 mg/mL. The median MBIC value for all tested isolates was 0.07 mg/mL. Of note, after incubation with SeNPs, three isolates increased their biofilm-producing capacity compared to the positive control containing untreated isolate. It is also noteworthy that the MBIC values of the SeNPs were significantly higher than MIC values (*p* < 0.001). There was no significant difference between the presence of the tested biofilm-related genes and MBIC values (*p* > 0.05).

The median MIC values among non-biofilm producers, weak, and moderate producers were 0.03 mg/mL. Strong biofilm producers had doubled MIC values compared to other groups, however, without statistical significance (*p* > 0.05). On the other hand, the median MBIC values significantly correlated with the biofilm abilities of the tested CRAB groups (*p* < 0.001), as represented in Table 2.

XDR isolates showed significantly lower median MICs of SeNPs (0.03, range: 0.0015–1.25) compared to MDR isolates (0.07, range: 0.015–1.25). In biofilm-producing isolates, XDR had lower MBICs of SeNPs (0.07, range: 0.007–1.25) compared to MDR isolates (0.15, range: 0.015–1.25), but without statistical significance (*p* > 0.05).

## 4. Discussion

Carbapenem-resistant and biofilm-producing *A. baumannii* pose a significant challenge to clinicians due to the increased incidence of hospital-acquired infections with high morbidity and mortality rates [1,27]. As expected, the vast majority of the isolates tested in the present study were recovered from the lower respiratory tract specimens (34.2%), blood (31.5%), and skin and soft tissue (23.4%), which is in line with previous reports [8,15]. Among the tested CR*AB* isolates, *bla*_OXA-23_ was the most frequent carbapenemase-encoding gene (36.4%), followed by *bla*_OXA-24_ (29.9%). Similar results have also been reported in countries in the same region and around the world [8,28,29,30,31]. One third of the CRAB isolates exhibited MDR phenotype, while almost 80% of *A. baumannii* presented XDR phenotype and 2.7% were classified as PDR. Polymyxins and tigecycline were the only antimicrobial options for XDR *A. baumannii.* However, the detection of three colistin-resistant isolates in this study is worrisome. Although still uncommon, *A. baumannii* resistance to colistin has been reported in different geographic regions [30,32,33]. Currently, for PDR *A. baumannii*, limited therapeutic options exist and are usually reflected in combination therapies of two or more antibiotics [34].

In the present study, the vast majority of the tested isolates were biofilm producers (91.9%). Indeed, *A. baumannii* is well known for its high capacity for biofilm production, which helps clinical isolates to survive in harsh environments and resist various antimicrobial attacks [35]. Interestingly, in the current study, all invasive isolates were biofilm producers. However, a slightly higher proportion of non-invasive isolates were strong biofilm producers, compared to invasive CRABs. Therefore, a clear relation between biofilm ability and invasiveness was not detected. Similar findings have already been reported [36]. According to the predominant literature data, non-invasive isolates are usually better biofilm producers compared to invasive isolates [37,38]. It is likely that non-invasive strains need to attach firmly to the devices or human cells to establish infections, which directly leads to better biofilm-production ability. However, bacterial colonisation and biofilm formation of indwelling devices, especially intravascular and intracardiac catheters, urethral catheters, and ostomy devices and tubes, might be the source of sepsis, which could partially explain the fact that all *A. baumannii* isolated from blood and peritoneal specimens in the current study were biofilm producers. Nevertheless, the majority of them were moderate, followed by strong and weak biofilm producers. Moreover, the obtained results might also be affected by a low sample size of invasive isolates (*n* = 44). The most prevalent gene among biofilm-producing isolates was *csuE* (78.6%), followed by *ompA* (71.4%) and *bap* (57.1%). Similar results were reported by Sung et al., who found that *csuE* was the most prevalent gene associated with biofilm formation in clinical isolates of *A. baumannii* (93.8%), followed by *bap* (75.0%) and *ompA* (68.8%) [39]. Furthermore, literature data indicated that *Acinetobacter* spp. able to produce biofilm harboured at least one of the following virulence genes: *bap*, *ompA*, *epsA, csuE,* and *bfmS* [40]. 

Although *csuE* was the most prevalent gene found in this study, we showed that *A. baumannii* lacking *csuE* gene might also exhibit biofilm-formation ability. This finding is in concordance with Moon et al., who confirmed that biofilm production in *A. baumannii* was not *csu*-dependent [41]. Similarly, other authors reported that the *csu* gene was important for biofilm production and was more associated with strong biofilm producers than with weak or non-producers, but was not crucial for biofilm capacity [42,43]. Likewise, our results show that the *csuE* gene is detected more often in moderate (79.4%) and strong (80%) biofilm producers, which is in agreement with the fact that it codes proteins necessary not only for the adherence and initiation of biofilm formation but also for better production of biofilm mass and matrix quantity [44].

In the present study, the prevalence of the *bap* gene among biofilm producers (57.1%) was similar to results found in Thailand (48%) [45] and lower than that obtained in Korea (85.7%) [39]. In this study, the *bap* gene was detected in 36.4%, 53%, and 72% of weak, moderate, and strong biofilm producers, respectively. This has previously been confirmed in studies that demonstrated the role of the Bap protein in biofilm maturation rather than in the initial stage of adherence [46]. Authors from Iran, Thailand, and Korea found *ompA* in 81%, 84.4%, and 68.8% of isolates, respectively [38,45,47], which corresponds to our finding (68.8%).

Besides biofilm, the long-term survival of *A. baumannii* in adverse environments is due, in part, to its high disinfectant tolerance, which is a prerequisite for *A. baumannii* cross-infection and ongoing intensive care units’ outbreaks. Appropriate applications of 70% ethanol and 0.5% NaOCl could eliminate the sources of pathogens and interrupt their transmission, thus reducing the prevalence of hospital-acquired infections. The most common applications are the disinfection of abiotic environmental surfaces (e.g., working surfaces, floors), equipment, and furniture, including door handles. Furthermore, they can be used for biotic surfaces, such as for skin disinfection before peripheral catheter insertion, especially in patients with contraindications to chlorhexidine [48]. However, inappropriate exposure to disinfectants, such as the use of sub-inhibitory concentrations, may result in resistance to some antibiotics [49]. Additionally, strong and moderate biofilm-producing isolates need higher concentrations (MIC and MBIC) of disinfectant to kill bacteria. Studies addressing the emergence of *A. baumannii* resistant to antiseptics and disinfectants are extremely rare, or the number of isolates tested is insufficient. Therefore, this study evaluated the eradication potential of the main disinfectants used in Serbian hospitals, 70% ethyl alcohol, several hand sanitiser ingredients, and 0.5% bleach, on mature biofilms of a significant number of CRAB (*n* = 102). Disinfectants must be applied within the specified dwell time in order to effectively kill planktonic bacteria. Thus, ethanol typically has a contact time of 1 min [50], while bleach is a disinfectant at 1 min, but sterilises at 4 min, and common hospital practice recommends using a contact time of 4 min [51]. To eradicate the mature biofilm of MDR *A. baumannii*, we extended the recommended dwell time of the tested microbicides. The obtained results showed that 0.5% NaOCl displayed significantly better biofilm eradication potential compared to 70% ethanol, regardless of the exposure time. Similar results were found by other researchers [52], confirming that ethanol insufficiently disrupts polysaccharides in the biofilm matrix, while NaOCl damaged both the bacteria embedded in the biofilm and the biofilm matrix itself [53]. Therefore, ethanol should be restricted to hand disinfection, whereas NaOCl is more suitable for surfaces where mature biofilm is expected.

With the advent of innovative nanotechnology, SeNPs emerged as promising agents for biomedical uses due to their low toxicity, degradability, and high bioavailability. In the current study, SeNPs showed antibacterial activity against all tested CRABs, with MICs lower than 0.07 mg/mL for two thirds of the isolates. Accordingly, previously published articles strongly suggested the wide-spectrum antibacterial activity of SeNPs against a variety of microbes, including Gram-negative bacteria, such as *E. coli*, *P. aeruginosa*, and *A. baumannii* [54,55,56]. However, to the best of our knowledge, this is the first study evaluating the antibiofilm efficacy of SeNPs against CRABs with confirmed biofilm-producing capacity. A total of 90% of biofilm producers displayed an MBIC of SeNPs lower than 0.15 mg/mL, suggesting that SeNPs might be used both as a promising therapeutic agent and for medical device coating to serve as an alternative approach for the prevention of biofilm-related infections. A single report on the antibiofilm activity of SeNPs against six isolates of *A. baumannii* demonstrated a substantial reduction in biofilm mass [57]. In this study, a substantial proportion of the tested CRABs (42%) displayed the same MIC and MBIC values. Such results might suggest that SeNPs hinder biofilm formation, primarily by growth inhibition of planktonic bacteria, population density reduction, or disruption of quorum-sensing gene-regulation, thus preventing the initiation of biofilm production. Several reports confirmed that increased Se ions could not only disrupt cell walls but also destroy cell membrane integrity, damaging intracellular homeostasis and microbial dysfunction, thereby leading to microbial cell death [58]. However, the results of a previously published study on *E. coli* confirmed that SeNPs are able to reduce exopolysaccharide synthesis, inhibit biofilm formation, and eradicate mature biofilms [56]. A number of mechanisms may be involved in the prevention of biofilm formation by SeNPs, including inhibiting quorum sensing and depleting the overall number of bacteria. Nevertheless, bacteria can develop different resistance mechanisms to SeNPs. All suggested SeNP mechanisms of action are mediated by penetration of SeNP into the bacterial cell, which may be compromised by induced changes to its compositions. Thus, Hachicho et al. reported that *Pseudomonas putida* was able to change the conformation of the unsaturated fatty acids present in its membrane [59]. This change in conformation produces a change in the fluidity of the membrane, which makes it less permeable and prevents the passage of nanoparticles and ions [59]. It has been shown that bacteria have a capacity to increase the expression of various efflux pumps and to up-regulate antioxidant enzymes, changing the expression of several transport and stress-related genes [60]. Therefore, the widespread use of metal and metal oxide nanoparticles seems to stimulate the co-selection and co-expression of antibiotic-resistance genes. Finally, bacteria are also capable of altering their morphology, resulting in a lower bactericidal effect of nanoparticles. For example, Zhang et al. reported that *E. coli* was able to develop resistance to NPs by acquiring a smaller and more oval shape [61]. However, to the best of our knowledge, the resistance mechanisms of *A. baumannii* to SeNPs have not been completely elucidated.

In addition, further studies evaluating the synergistic effects of SeNPs and various antimicrobials for CRAB are needed, since some reports suggest that the combination of antibiotics with SeNPs may lead to stronger attacks on bacterial membranes, which facilitates the entry of both agents and subsequent additional antibacterial effects [14,62].

## 5. Conclusions

Together, our results suggest the need for a more extensive decontamination process when CRAB biofilms might be involved. Thus, while proper decontamination is important, a multi-disciplinary approach is essential when attempting to combat this challenging pathogen. Nanotechnology offers a ground-breaking platform to tackle MDR infections associated with biofilms. Thus, SeNPs exhibited remarkable properties that make them profitable non-antibiotic agents and biofilm inhibitors against some of the most challenging MDR bacteria, such as CRAB. Nevertheless, further analysis and in vivo studies are needed to determine the various aspects of employing SeNPs.

## Figures and Tables

**Figure 1 microorganisms-11-00171-f001:**
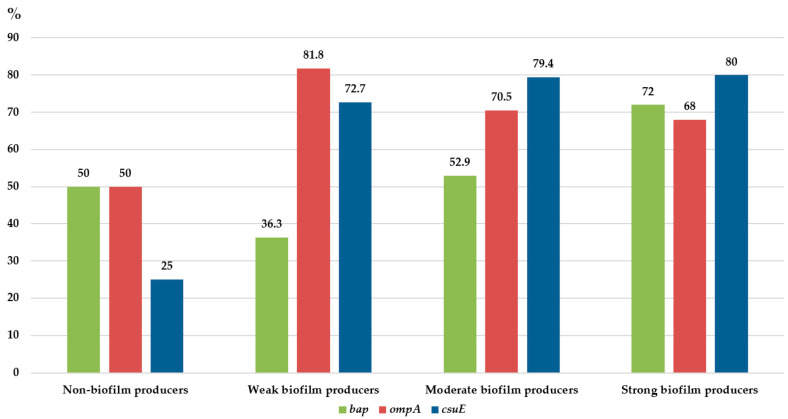
The percentage of gene-encoding biofilm-associated proteins among biofilm-producing and non-biofilm-producing carbapenem-resistant *Acinetobacter baumannii* (*n* = 111).

**Figure 2 microorganisms-11-00171-f002:**
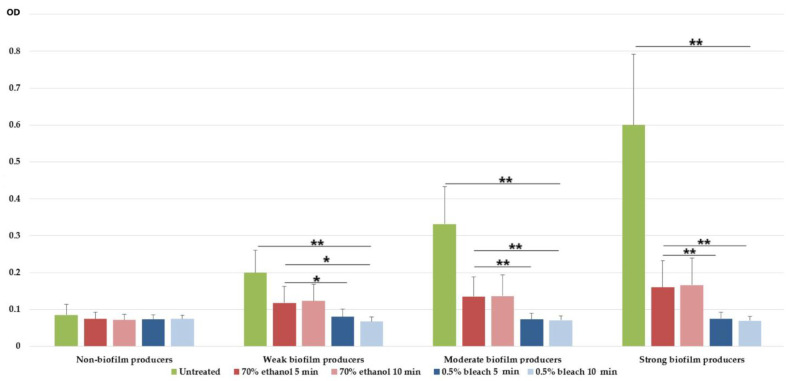
Eradication effect of 70% ethanol and 0.5% NaOCl on biofilm of CRAB strains. The effect was observed after 5 and 10 min of exposure for both substances. *—*p* < 0.05; **—*p* < 0.01. Biofilms are shown as mean ODs ± SD of all carbapenem-resistant *Acinetobacter baumannii* isolates within each biofilm capacity group.

**Figure 3 microorganisms-11-00171-f003:**
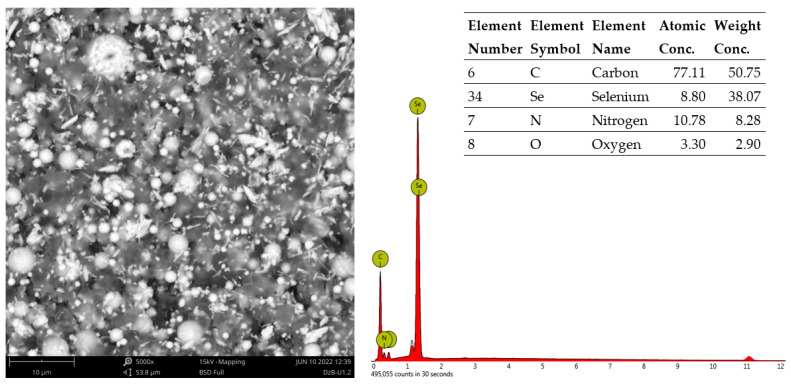
Scanning electron microscopy micrograph of selenium nanoparticles and energy-dispersive X-ray elemental analysis of synthesised sample of selenium nanoparticles.

**Table 1 microorganisms-11-00171-t001:** Biofilm-production capacities among 111 invasive and non-invasive carbapenem-resistant *Acinetobacter baumannii* (CRAB) isolates obtained from various clinical specimens.

Biofilm-Producing Capacity	Isolates N (%)					
	Non-Invasive CRAB (*n* = 67)			Invasive CRAB (*n* = 44)		
	Skin and Soft Tissue (*n* = 26)	Urine (*n* = 3)	Lower Respiratory Tract Samples (*n* = 38)	Peritoneal Fluid (*n* = 8)	CSF (*n* = 1)	Blood (*n* = 35)
Non-producers	4 (15.4%)	0 (0%)	5 (13.2%)	0 (0%)	0 (0%)	0 (0%)
Total	9 (13.4%)			0 (0%)		
Weak producers	4 (15.4%)	0 (0%)	1 (2.6%)	0 (0%)	0 (0%)	8 (22.9%)
Moderate producers	7 (26.9%)	2 (66.7%)	22 (57.8%)	0 (0%)	0 (0%)	20 (57.1%)
Strong producers	11 (42.3%)	1 (33.3%)	10 (26.4%)	8 (100%)	1 (100%)	7 (20%)
Total	58 (86.6%)			44 (100%)		

CRAB—carbapenem-resistant *Acinetobacter baumannii;* CSF—cerebrospinal fluid.

**Table 2 microorganisms-11-00171-t002:** Distribution of median minimum inhibitory concentration (MIC) and minimum biofilm inhibitory concentration (MBIC) values among non-producers, weak, moderate, and strong biofilm-producing isolates.

	Non-Producers (n = 9)	Weak Producers (n = 13)	Moderate Producers (n = 51)	Strong Producers (n = 38)
median MIC (mg/mL) [range]	0.03 [0.01–1.25]	0.03 [0.001–1.25]	0.03 [0.01–0.15]	0.07 [0.001–1.25]
median MBIC (mg/mL) [range]	0.00 [0.00–0.30]	0.03 [0.01–1.25] **	0.07 [0.01–0.60] **	0.15 [0.01–1.25] **

*** p* < 0.001.

## Data Availability

Not applicable.

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
