# Peer review of "Carbapenem-Resistant Acinetobacter baumannii: Biofilm-Associated Genes, Biofilm-Eradication Potential of Disinfectants, and Biofilm-Inhibitory Effects of Selenium Nanoparticles"

_microorganisms, 2023, doi:10.3390/microorganisms11010171_

Round 1

Reviewer 1 Report

General comments: 

In this manuscript, the authors have characterized the biofilm formation capacity of carbapenem-resistant Acinetobacter baumannii clinical isolates. And the authors have also demonstrated the inhibitory effects of selenium nanoparticles (SeNPs) on A. baumannii biofilms. Overall, this manuscript could provide a new clue to eradicate and inhibit A. baumannii biofilm formations; however, several issues should be clarified.

Specific comments:

1. In Table 1, what are the definitions of “invasive” and “non-invasive”?

2. Does Table 1 demonstrate that the biofilm formation capacity is unrelated to the invasiveness? Additional discussion is required about this point.

3. Figure 1 shows that clinical isolates obtained from blood samples grow at least weak biofilms and are not non-biofilm producers. This point seems interesting. Does this result suggest that some floating cells, dispersed from biofilms formed somewhere, were obtained? 

4. Concerning Figure 2, if the csuE gene get deleted in some clinical isolates, will these strains lose the biofilm formation capacity? 

5. In Table 2, some strains show resistance against SeNPs. What do the authors speculate about the resistant mechanisms? 

6. Do SeNPs possess eradication effects on mature biofilms? If so, this data would be informative to expand the clinical application of SeNPs.

7. Are there any synergistic effects of SeNPs with conventional antibiotics?

8. What are the clinical applications of 70% ethanol and 0.5% sodium hypochlorite against mature biofilm-related infections?

Reviewer 2 Report

The paper “Carbapenem-resistant Acinetobacter baumannii: biofilm-associated genes, biofilm eradication potential of disinfectants, and biofilm inhibitory effects of selenium nanoparticles”  presents a series of results on Acinetobacter baumanii, a pathogen of serious concern.

Though there are some interesting results, the paper presents a miscellaneous collection of data, showing that conceptualization and design of the work were not adequately carried out.

Overall, the authors claim that one of the aim of the study (lines 80-82)  is the analysis on the relationships between biofilm formation, biofilm-associated genes, and antimicrobial resistance. In my opinion the investigation failed to answer to the points so as to lead to original outputs, but it was limited to a screening of strains and a collection of data.

In the Results:

Line  210,  the invasiveness of the strains is not explained.

Fig 1 is redundant, and results could be inserted in Table 1.

The part which would deserve a deeper analysis is the use of selenium nanoparticles to control Acinetobacter biofilms. At the moment results appear very preliminary. The strains displayed the same MIC and MBIC values: so are SeNPs real antibiofilm compounds? What are their effect in terms of growth inhibition and what in terms of antibiofilm effect (quorum sensing, effect on genes…etc.)? A further detailed investigation is needed.

Also, the presentation of SeNP characterization (Fig. 4) is unsatisfactory (the SEM photograph is of low quality).

For the above reasons, I do not recommend the publication of the paper in the present form.

Round 2

Reviewer 1 Report

All the points raised in the first version of the manuscript have been properly addressed by the authors.

Reviewer 2 Report

The paper has been completely revised and authors have followed the Reviewer's suggestion, so that it can be published in  present form.